# Wildfire Smoke Exposure During Pregnancy: Consensus-Building to Co-Create a Community-Engaged Study

**DOI:** 10.3390/ijerph21111513

**Published:** 2024-11-14

**Authors:** Kelsie Young, Kim Alisa Brown, Lynda Crocker Daniel, Katherine Duarte, Diana Rohlman

**Affiliations:** 1Oregon Clinical & Translational Research Institute, Oregon Health & Science University, Portland, OR 97239, USA; 2Pacific Northwest Environmental Health Sciences Center, Oregon State University, Corvallis, OR 97331, USA; kim.brown@oregonstate.edu; 3Klamath County Public Health, Klamath Falls, OR 97603, USA

**Keywords:** community-engaged research, environmental health, perinatal health, consensus-building, wildfire smoke

## Abstract

Relative to other Oregon counties, Klamath County experiences worse air quality due to wildfire smoke, as well as elevated rates of infant mortality and low birthweight. Klamath County Public Health (KCPH) raised concerns that wildfire smoke is a contributor to poor infant health. Thus, we built a multidisciplinary team and designed a community-engaged research (CEnR) project to capture community and individual-level exposure to wildfire smoke contaminants, alongside perinatal health outcomes. Through partnerships, we identified 24 individuals across academic, public health, and community organizations that met five times over three months to develop a study design. We initially used a modified Delphi method, but adjusted our approach to find multidisciplinary areas of agreement across a highly diverse team. Our team used structured meetings, surveys, and iterative feedback to build consensus on a study design. KCPH and our community partners reviewed and approved all proposed activities to ensure community input was integrated. The resultant study, trialed in Klamath County, included the use of environmental, residential, and personal samplers and health surveys with a cohort of pregnant individuals during the wildfire season. We discuss the advantages and challenges of building a multidisciplinary CEnR study in a rural county disproportionately impacted by wildfire smoke and infant mortality.

## 1. Introduction

The northwestern region of the United States is experiencing longer and more severe wildfire seasons, with wildfire smoke impacting millions of people [1,2,3]. Exposure to wildfire smoke has been associated with increased mortality, as well as adverse respiratory and cardiovascular outcomes [4]. Much of this work has used a specific pollutant, PM_2.5_, as a proxy for exposure to wildfire smoke, given that PM_2.5_ is a major constituent of smoke [5]. Within the western United States, wildfires alone are estimated to account for up to 50% of all PM_2.5_ emissions during the past decade [6].

The impact of wildfire smoke exposure during pregnancy on birth outcomes and infant health has been a growing research area as fires become more common [7,8,9,10,11,12]. A growing body of preliminary research suggests that prenatal exposure to wildfire smoke is associated with preterm birth [7,9,11] and low birth weight [7,8,10,12]. Preterm birth is defined as infants born before 37 weeks of gestational age, while low birth weight (LBW) is defined as an infant born weighing less than 2500 g (5 lbs, 8 oz). Preterm birth and LBW are associated with a greater risk for infant mortality, developmental disabilities, obesity, diabetes, and other chronic conditions [13].

In Klamath County, Oregon, residents have been experiencing more severe wildfire seasons since 2012 and have more air quality days rated as “unhealthy for sensitive groups” relative to other counties in Oregon [14]. From 2015–2022, there were 117 days where air quality rose above 100 on the Air Quality Index, rated as “unhealthy for sensitive groups”, or an average of 14.6 days/year, positioning Klamath County as having the worst air quality due to wildfires in Oregon [14]. In contrast, the second worst region was Jackson County, which had an average of 12.7 days/year at or exceeding “unhealthy for sensitive groups” [14].

Klamath County has a long-standing concern regarding infant health. From 1995–2021, Klamath County’s infant mortality rate frequently surpassed the State of Oregon, with an average of 7.7 deaths per 1000 live births, versus an average of 5.2 deaths per 1000 live births across Oregon [15,16]. The rate of infants born with a low birth weight (LBW) has also been relatively high. From 2009–2021, Klamath County’s percentage of LBW infants consistently exceeded the State of Oregon, with an average of 8.5%, versus an average of 6.5% across Oregon; the highest observed rate of LBW infants in Klamath County reached 10.7% in 2021 [15,16].

Klamath County Public Health (KCPH) has been spearheading efforts to address these adverse infant outcomes, and leads a workgroup comprised of community-based partners—Trends on Thriving (TOTs)—to identify potential interventions to lower the rate of infant mortality, such as smoking cessation and prenatal health care. In early 2021, the workgroup expressed concerns regarding wildfire smoke exposure during pregnancy. To address these concerns, KCPH partnered with researchers at Oregon State University (OSU) to build a team and design a study to assess perinatal exposure to wildfire smoke. The team was charged with evaluating the following questions: “What data and health endpoints are most useful? How can this study best address the needs of KCPH and similar communities? How can the study address research interests?” In addressing these questions, the multidisciplinary team developed strategies for integrating research and community priorities and collaboratively identifying hypotheses, study design, data collection tools and methodologies, and health outcomes to develop a pilot study in Klamath County.

## 2. Methods

### 2.1. Project Initiation and KCPH–OSU Partnership

In 2021, a TOTs member and Community Research Liaison with the Oregon Clinical and Translational Research Institute (OCTRI) Community Research Hub reached out to the OSU Pacific Northwest Environmental Health Sciences Center (PNW EHSC) to discuss KCPH concerns regarding wildfire smoke exposure during pregnancy.

The PNW EHSC includes a Community Engagement Core (CEC), with the mission of facilitating community-engaged research (CEnR) between academics, communities, decision-makers, and health agencies to co-produce relevant, timely, and actionable environmental health decision-making. In May of 2021, the CEC, KCPH, and the Community Research Liaison met to discuss the Klamath county data, current research on wildfire smoke exposures during pregnancy, and the interests and goals of KCPH and the TOTs group (Figure 1). Given these common interests, this group formed the core community-academic research team (CCART) with the goal of expanding the team expertise to co-create a community-engaged pilot project that would support larger, longitudinal studies designed to assess whether the preterm rates of birth in Klamath are attributable, at least in part, to wildfire smoke. The team met over a period of six months (Figure 1). Funding was provided by the PNW EHSC for this team-building exercise.

### 2.2. Building a Multidisciplinary Team

The CCART disseminated an email to 48 individuals with appropriate expertise and/or interest in the project areas (perinatal health, toxicology, epidemiology, and wildfire smoke) (Figure 2). An explanation of the project was provided, and recipients were encouraged to forward the email to additional individuals or organizations that may have had an interest in the project. In this way, at least eight additional potential partners received an invitation; four opted to participate in the project.

Twenty-two people in addition to the CCART (*n* = 6) attended the project kickoff meeting held online in November 2021 (Figure 1), representing 13 academic, public health, and community organizations in the following disciplines: toxicology, perinatal health, public health, community engagement, air quality, statistics, and chemistry (Table 1). In sum, 28 individuals continued to participate in the project.

### 2.3. Consensus Building

Initially, we utilized a modified Delphi Method [17]. We held virtual meetings, followed by surveys disseminated through Qualtrics^TM^ (Provo, UT, USA). Meetings were led by a facilitator and provided a forum for controlled feedback following a review of the statistical representation of group survey responses (Table 2).

However, meeting attendance and survey responses were highly variable. As a result, we shifted from the structured approach of the Delphi method to a more informal, qualitative approach. We continued to meet with the team virtually and distributed the surveys as originally planned, but the representation of the results was used solely to initiate conversation within the group. With those present in the meeting, we came to a consensus. Detailed notes were distributed after each meeting for all members, including those that were unable to attend, to provide additional feedback as necessary. For the final meeting, the CCART presented the final study design, and approval was asked of meeting attendees; opportunities for dissent were also provided via follow-up emails to the full 28-member team. In the absence of disagreement, the study design was finalized.

### 2.4. Meeting Design and Structure

Team members were located across Oregon and in Washington, necessitating virtual meetings. Table 2 describes the meeting format and topics presented at each meeting. Briefly, elements of a study design were presented at each meeting for discussion. A single team member facilitated each meeting. The meetings began with an overview of the project and a review of any applicable survey responses. All meetings were audio-recorded, and note-takers were present. Notes were confirmed with the audio recording to generate a final meeting summary. Each meeting was typically one-and-a-half hours. Honoraria were offered to community and academic participants as part of the team building process. A study website with general information about the project and contact details was developed and meeting recordings and notes were posted on the website immediately after meetings 2–5.

### 2.5. Survey Development

As shown in Figure 1 and further described in Table 2, three surveys were disseminated to the team via Qualtrics^TM^. Generally, surveys consisted of 10 questions (range 6–16 questions) and were designed to be completed in under 20 min. Question types included Likert scales, ranking, and free-text questions. Given the diversity of expertise represented in the team, surveys were designed to be broadly accessible in terms of interface and access to information. Surveys were accessible by computer, tablet, or smartphone. Scientific terms were hyperlinked to plain language descriptions from evidence-based organizations (e.g., descriptions of PM_2.5_ linked to a resource from the United States Environmental Protection Agency) (Appendix A).

Surveys were administered via email with an anonymous link. While the team strove for a one-week response time, in practice, team members were asked to complete the online surveys within an average of three days of initial dissemination given the project timeline. Email reminders were sent out a day prior to scheduled meetings.

### 2.6. Data Analysis

Responses to open-ended survey questions were collated into a Word document and reviewed by the CCART. Areas of consensus, as assessed by discussions in team meetings and from written responses, were used to make decisions, e.g., selection of a residential PM_2.5_ air monitor. Areas of disagreement or lack of consensus were brought up for further discussion in team meetings.

Several survey questions utilized a five-point Likert scale. These data were evaluated using a weighted-rank analysis where each rank is multiplied by the number of respondents (N) who selected each ranking. These rank–respondent products are summed, and the sum is divided by the total number of respondents using the following formula:$$\text{Weighted-rank}=\left[\left({Rank}_{1}\times N_{1}\right)+\left({Rank}_{2}\times N_{2}\right)+...+\left({Rank}_{n}+N_{n}\right)\right]/(N_{1}+N_{2}+...+N_{n}).$$

Between the meeting notes and survey responses, the CCART made informed decisions on study elements. These decisions also reflected the delicate balance between academic interests and community concerns (Appendix A). These decisions, as well as the underlying survey responses and meeting notes, were presented at subsequent meetings to inform further discussion. This iterative evaluation ensured the full team understood how their feedback was collected, how it was evaluated, and how it informed decisions.

## 3. Results

### 3.1. Transitioning Away from the Modified Delphi Method

Despite strong interest from the team, survey response rates were on average 28% (Table 2). One reason for the low responsivity was identified in a meeting, wherein community partners noted they felt they lacked the scientific expertise to complete the surveys, despite explanatory information embedded in the surveys. Additionally, attendance from team members was highly variable, given busy and conflicting schedules. Therefore, results from the surveys were not used as evidence of consensus. Rather, the survey data were presented at each subsequent meeting to inform discussion. Data were presented in tabular and graphic form, and direct quotes from open-text responses were included as applicable. In all meetings, the CCART described the feedback that was received from prior meetings and surveys and how the data were then analyzed to identify areas of consensus. Meeting summaries were distributed and revisited at subsequent meetings to ensure all participants had multiple opportunities to provide feedback. The consensus process first identified the various stages of a study and the elements of a study (Figure 3). The team engaged in scientific questioning around the following elements: pollutants of interest, data collection methodologies and tools, health outcomes of interest, and the overall study design. A full description of all features that were evaluated and then selected by the team, with rationale, are described in Appendix A and briefly summarized below.

### 3.2. Building Consensus—Pollutants of Interest

While PM_2.5_ was the major pollutant of interest, given its abundance in wildfire smoke [5], the team evinced interest in evaluating exposure to additional pollutants. The CCART consulted the peer-reviewed literature to identify (1) pollutants found in wildfire smoke and (2) pollutants associated with adverse birth outcomes. The results were presented to the team in a survey and then in a meeting to initiate discussion, encourage questions, and solicit additional input. While carbon monoxide (CO) was not initially listed, an academic partner identified it during the meeting, and the group agreed it should be monitored given that CO levels may increase due to wildfires, and CO monitors are cost-effective and easy to use. A consensus was reached to measure exposure to the following pollutants of interest: PM_2.5_, carbon monoxide (CO), 63 polycyclic aromatic hydrocarbons (PAHs [19]), and an additional 1530 semi-volatile and volatile organic compounds [20] (SVOCs/VOCs) given the analytic capabilities available to the team and the PNW EHSC (Figure 4, Appendix A).

### 3.3. Building Consensus—Data Collection Methodologies & Tools

Using the research question identified by the CCART, the team was asked to provide input on sampling options. Direct (sampling on participants), indirect (sampling the participant’s environment), and biomonitoring (sampling participant blood, urine, hair, etc.) sampling options were presented to the team. The team quickly came to agree that both direct and indirect sampling methodologies were appropriate. For example, PM_2.5_ monitors were selected for direct and indirect sampling (Appendix A), and personal and environmental passive samplers were selected to characterize PAHs and other SVOCs and VOCs. One team member felt strongly about collecting biological samples, e.g., blood, urine, and placental samples (indicated through a survey response). However, during the subsequent meeting, community-based organizations discouraged using invasive sampling such as blood or urine, as their patients frequently express concern that such samples will be used for drug-testing and are therefore reluctant to participate in studies using biological data collection. Biological samples were not included in the study design. However, the team agreed that future sampling campaigns may include the option for providing biological samples.

In total, two residential samplers (PM_2.5_ and CO monitors), one personal sampler (silicone wristband—measures SVOCs/VOCs [20,21]), and three environmental samplers (PM_2.5_, CO monitor, and a passive sampler for measuring SVOCs/VOCs) were selected to measure the exposure levels to ambient and indoor air contaminated with the wildfire pollutants PM_2.5_, CO, PAHs, and additional SVOCs/VOCs (see Figure 4). The environmental samplers were co-located with the two state-level air monitoring stations managed by KCPH.

The team further recommended that the study should be supplemented with surveys, enabling participants to record additional information about their potential exposures and their health. While considered an excellent source of data, electronic health records were not included, given time constraints and the difficulty of accessing these records across multiple health organizations. Instead, several surveys were disseminated to assess health outcomes, using pre-existing and/or validated surveys where available (Appendix A).

### 3.4. Building Consensus—Health Outcomes

We addressed the health outcomes last, as KCPH was clear on their desire to assess perinatal health, with an emphasis on infant mortality and low birth weight. However, we wanted to ensure the team had a chance to consider additional health outcomes.

The CCART identified additional health outcomes representative of perinatal health, with additional inclusion of metrics that have been, or potentially are, negatively associated with exposure to wildfire smoke (Appendix A). Furthermore, the team recommended daily health surveys to capture the respiratory health of the pregnant person, given the well-established impact of wildfire smoke on respiratory health [22,23,24].

The team expressed interest in evaluating health beyond the perinatal period, for example tracking health throughout childhood, yet agreed this was beyond the scope of a pilot project. Given the prior difficulties identified with integrating electronic health records (Appendix A), the team instead decided to collect health outcome data via three strategically timed surveys: a health history survey administered upon enrollment; daily health surveys during the study, and a birth outcome survey administered shortly after birth, which would collect infant health outcomes (Appendix A). An optional feasibility survey was additionally included to assess self-reported compliance with the study protocol and samplers. These surveys utilized existing survey tools and selected validated tools where possible.

Team members suggested additional tools for directly measuring health, such as self-administered spirometry or monitors for measuring the amount of nitric oxide in exhaled breath; both can capture an assessment of respiratory health. However, when evaluating these options with the team, when compared to the other data collection tools, these were deemed difficult to use, and there was insufficient evidence to indicate the tools would capture biological changes as a result of acute exposure to wildfire smoke. For example, while lung function has been seen to decline following exposure to wildfire smoke, this is a long-term effect, rather than one experienced acutely during or immediately after wildfire smoke exposure [25].

### 3.5. Building Consensus—Study Design

Given the exposure of interest (wildfire smoke) a repeat-measures design was proposed, wherein participants would sample prior to wildfire season (nominally April–June), and then sample again during a wildfire (anticipated to occur July–September). Given the gestational period and the sampling period of interest, this limited enrollment to individuals that were at or under 16 weeks’ gestation, with no underlying health concerns. The sampling period was limited to four days, given logistic considerations for the personal PM_2.5_ sampler, which had a limited battery life. This sampler was later discarded due to a complicated user interface and setup (Appendix A) but the sampling period was maintained given that wildfire smoke patterns can shift quickly. The total time for a participant to complete the project, defined as time spent undergoing recruitment and enrollment, setting up monitors, and completing surveys, was just over 2 h. Given budgetary constraints, the pilot project was designed to sample twenty individuals and then analyze data from five, based on severity of exposure to wildfire smoke. The team remained engaged past the end of the meeting schedule, with 28 team members approving the subsequent grant application.

### 3.6. Overview of Pilot Project

Our study enrolled seven participants, and six completed both sampling periods that occurred in September and October of 2022. In total, 19 individuals initiated the eligibility screener; eight were either ineligible or did not complete the screener; four were unable to schedule phone interviews. Funding from the study was used to develop an educational infographic, Wildfire Smoke and your Baby, in English [26] and Spanish [27], to provide timely information before the upcoming wildfire season to build awareness and help reduce exposures. Feedback from the study participants indicated that the study protocol was well-tolerated, and the equipment was easy to use. Study updates were released in January 2022 and February 2023, and the team shared individual and aggregate results to participants in August 2023 regarding their exposures to PM_2.5_ and CO during the study. Participants also received individual and aggregate results from the silicone wristbands for their data related to exposure to 63 PAHs (June 2024) and over 1500 chemicals (September, 2024). The OSU team and the CCART team will meet to review all the aggregate results in Fall 2024.

## 4. Discussion

Following a review of the successes, challenges, and mitigation strategies utilized during this project, we developed a list of ‘lessons learned’ for future CEnR projects, particularly those with multidisciplinary teams (Table 3). The inclusion of local knowledge and expertise from community partners was crucial in ensuring that our project design was aligned with community interests and expectations. Furthermore, it was essential that our process equally weighed academic and community expertise. Our team building process brought together diverse expertise and skill sets across academics, practitioners, and communities. We were challenged by varying vocabularies as well as by differences between regulatory, academic, and community perspectives and the desire to balance pragmatic approaches relative to scientific inquiry. Our goals with using a meeting facilitator were to balance the power dynamics to ensure all team members had an opportunity to contribute their expertise and to enhance communication during meetings (e.g., minimize jargon and explain acronyms as needed). A benefit of this approach was the embedded opportunity for iterative evaluation, enabling us to pivot to different strategies as needed to appropriately gather feedback and obtain consensus. A tangential strategy to ensure a robust team was the development of user-friendly and accessible surveys and meeting slides at every stage. This approach was designed to ensure that materials were accessible but also transparent, as we attempted to mitigate knowledge gaps by using plain language definitions for technical terms. Furthermore, we spent substantial time pre-planning meetings, reviewing previous input, and developing the above strategies with our CCART to make the conversations in our team meetings inclusive and accessible. However, we quickly found that the structured format of the modified Delphi method was a poor fit, so we switched to a more informal strategy to ensure equivalent feedback from our diverse partners.

We acknowledge that a compressed timeline (3 months; Figure 1) and the level of input required in between the meetings made it more challenging for some members to be involved in all aspects of the project design process and it was important to provide at least one week for a final proposal review. Communication and access to resources in between the meetings were critical to creating an inclusive and aligned team, but even with these efforts, this project would have benefited from a longer timeline.

## 5. Conclusions

With a community-engaged approach, we were able to leverage the knowledge and experience of experts representing 13 academic, public health, and community organizations over a period of three months. Utilizing a model of consensus building with this multidisciplinary team resulted in a comprehensive study design that was built upon diverse expertise to address the community’s environmental health concerns and a successfully funded pilot study. Presenting the results of the surveys during online meetings and engaging all participants in discussion and transparent decision-making also helped us leverage community knowledge and lived experience within the group and quickly identify areas that were not a good fit for the community.

While the Delphi method is designed to be useful for gaining expert insight and weighing multiple opinions, the response rate for surveys was low, and meeting attendance for both academic and community team members was inconsistent. Additionally, the study design process had external time pressure with a time-sensitive grant deadline. This left just under 3 months to build a team, solidify a study design, and submit a grant proposal. Additional time to participate would likely have benefited community partners. Despite these challenges, our team identified a more flexible approach for collecting, visualizing, and then discussing study elements with a multidisciplinary team using community-engaged strategies, resulting in a successful pilot study. The lessons learned herein may serve useful for future team-building efforts.

## Figures and Tables

**Figure 1 ijerph-21-01513-f001:**
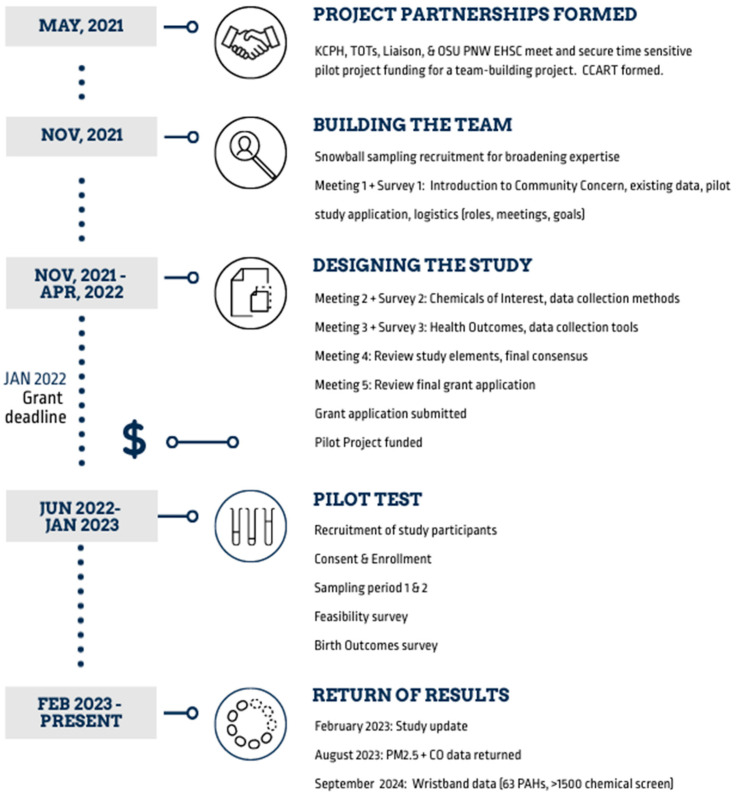
Timeline of the team-building project and subsequent research study.

**Figure 2 ijerph-21-01513-f002:**
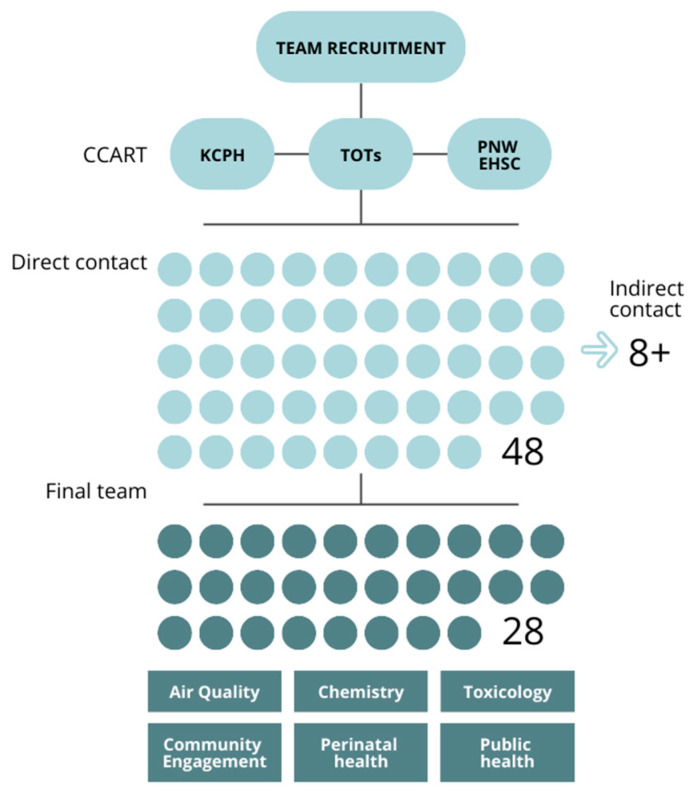
Strategies for building a team. A mix of direct emails and indirect email (contacts were encouraged to forward emails on to others with interest and expertise) methodologies were used. While the total number is unknown, indirect email recruitment resulted in at least eight individuals receiving the invitation. Upon being contacted, a total of 28 individuals participated in at least one meeting, to comprise the final team.

**Figure 3 ijerph-21-01513-f003:**
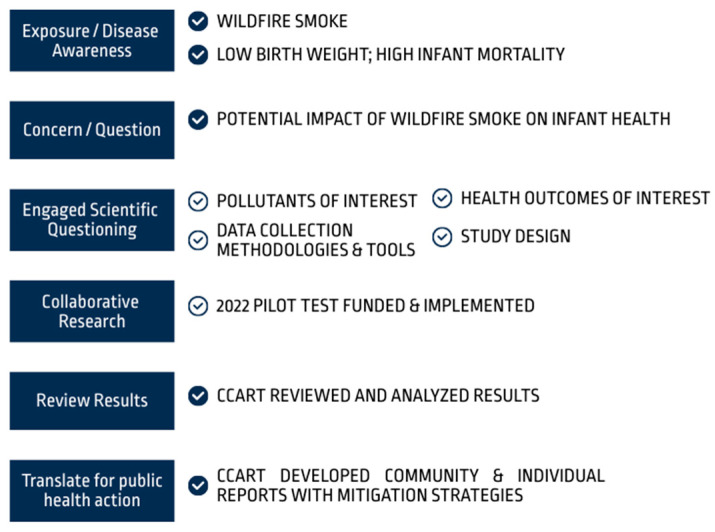
Using a model of Community-Engaged Research and Citizen Science (O’Fallon, 2015 [18]), the team collaborated to move through the stages of the research process. The first two stages (exposure/disease awareness and concern) were initially determined by KCPH. Filled checkmarks next to research stages represent decisions made by the CCART; open checkmarks represent decisions that were made in consensus with the larger team.

**Figure 4 ijerph-21-01513-f004:**
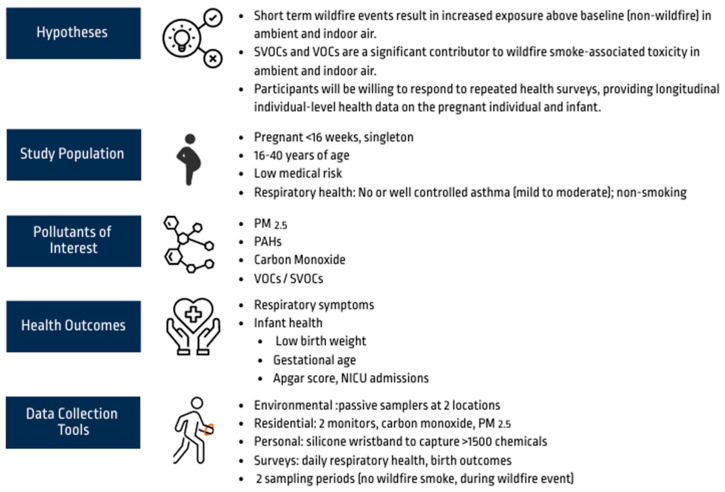
Summary of the pilot project designed by the CCART and larger team. The team provided input on all aspects of the study.

**Table 1 ijerph-21-01513-t001:** Description of the multidisciplinary research team. Members represented 13 academic, public health, and community organizations. *Collaborators self-identified their expertise, and many selected more than one discipline. ^1^ Land-grant colleges or universities are those that have been designed to receive benefits through the Morrill and Hatch Acts, with the mission of teaching and conducting agricultural studies and research. ^2^ Coordinated Care organizations are networks of health care providers that coordinate and deliver care to people covered by the Oregon Health Plan or Medicaid, in the State of Oregon. ^3^ Pediatric Environmental Health Specialty Units are national units bringing together experts in child health issues associated with environmental exposures.

Expertise/Discipline	Organizational Affiliation	Number of Experts *
Air Quality	Public polytechnic and research university; State health department; County health department; Land-grant university ^1^	7
Chemistry	Public polytechnic and research university; State health department; Land-grant university	3
Perinatal Health	Public polytechnic and research university; Land-grant university; County public health department; Government nutrition program; Non-profit organization; Coordinated care organization (CCO) ^2^; Family birth center; Pediatric Environmental Health Specialty Unit ^3^	11
Community Engagement	Public polytechnic and research university; Tribal health services; Non-profit organization; Public research university; Coordinated care organization (CCO)	8
Public Health	County public health department; State health department; Tribal health services; Government nutrition program; Public research university; Land-grant university	15
Toxicology	State health department; Land-grant university	3
Statistics	Land-grant university; State health department	5

**Table 2 ijerph-21-01513-t002:** Description of the five virtual team building meetings, estimated attendance, and associated survey response rates. ^1^ Attendance was not strictly tracked, as some members joined for the first half of the meeting, while others joined for the last half of the meeting. The numbers below are reflective of the number of attendees at the beginning of each meeting. n/a = not applicable.

Meeting	Meeting Topic	~Attendees ^1^	Survey
1	Introductions, community concern, existing data, research process, project goals, timeline, roles	27	16 questions 17 respondents/39 sent (44% response rate)
2	Chemicals of interest and data collection methods	12	6 questions 3 respondents/33 sent (10% response rate)
3	Health outcomes and data collection tools	12	7 questions 10 respondents/33 sent (30% response rate)
4	Review study elements and final consensus	13	n/a
5	Review final grant application	13	n/a

**Table 3 ijerph-21-01513-t003:** Summary of lessons learned following review of challenges and mitigations of our team building and project design process.

Lessons Learned
Inclusion of community voice and expertise is essential Utilize strategies that weigh academic and community expertise equally Continually evaluate strategies for gathering input, pivot as needed Ensure relevance, transparency, and accessible materials Plan for team debriefs and substantial meeting facilitation Incorporate longer timelines for multidisciplinary teams

## Data Availability

No new data were created or analyzed in this study. Data sharing is not applicable to this article.

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
