# Peer review of "Wildfire Smoke Exposure During Pregnancy: Consensus-Building to Co-Create a Community-Engaged Study"

_ijerph, 2024, doi:10.3390/ijerph21111513_

Round 1

Reviewer 1 Report

Comments and Suggestions for Authors

Overall this is an interesting study that documents a community engagement strategy. 

Figure 2: The "48", "28" and "8" part is a bit confusing as the dots do not represent the numbers. Is it possible to have open dots for invited and filled dots for attended or something?

Table 1: The identities of some of the entities are not necessarily clear. Consider providing a more detailed table in the supplemental information identifying participating team members, or explaining what land-grant and other types of universities as well as entities like PEHSUs. 

Section 2.3 consensus building: it sounds like stakeholders were disengaged with the initial meeting format, but it is not clear when the format changed (which meetings) as only the differences in the last meeting were described. It would help for the initial meeting structure and adaptations to future meetings to be more clearly described. 

Survey response numbers were low. SUmmaries were discussed in meetings, but it is acknowledged that with time pressure and the number of team members it was not always possible to attend. Is it possible to summarize the number of team members who attended each meeting as well as those who completed surveys?

Discussion:

There are some parts of the discussion where links between the data collected and the conclusions made are not transparent to the reader. The meeting facilitator and goals for using one came up for the first time in the discussion. It is not clear that their function was met, with challenges that included vocabulary, differential views and community members not feeling comfortable with surveys. The iterative evaluation and adaptation process could be better described in the methods and even the results, making that part of the conclusion clearer; this was a purpose that is commented on but definitely could not be replicated with this level of detail. Features that were evaluated, and how changes were designed to meet the needs, would be helpful information. 

Lessons learned: One of the goals of the project was to include community voice, so it isn't really a lesson. The community preferred not to collect biosamples, but it is not clear whether the participants in the small pilot study were asked about the study features that were/were not included, to see whether different opinions at the committee level played out in reality. The second lesson also is more of a project goal than a lesson. The other 4 make sense within the context of the project. 

Reviewer 2 Report

Comments and Suggestions for Authors

Thank you for your work captured in this commentary.

This was a undoubtably a challenging exercise and you provide good detail in your summary.

One notable exception was summary statistics from surveys. This would be useful to further underscoring your inferences and framing the challenges of the study.

Additional comments:

-            This commentary describes assessment of wildfire smoke contaminants in Klamath County, initially with modified Delphi, which was adjusted to further capture multidisciplinary variations, during the wildfire season

Major issues (might include problems with the study’s methodology,
techniques, analyses, missing controls or other serious flaws)

-            There are no major issues or serious flaws with the methodology, techniques or analyses approach offered in this commentary. I agree with the methodologic decision to switch away from the framework of the modified Delphi method.

Minor issues (might include tables or figures that are difficult to read,
parts that need more explanation, and suggestions to delete unnecessary
text)

-            While this commentary is not to describe results, it is of value to the reader to see some preliminary results from the surveys carried out and interviews where possible. Especially as Section 2.6 describes a Data analysis approach. The authors should include preliminary summary estimates in an attached table either within the commentary or as a supplementary file.

-            In addition, the authors briefly touch on issues with limited time for data collection, also impacting/resulting in a low survey response rate and inconsistent participation by study participants and respondents. The authors should comment on the possible methodologic and inference implications of these limitations on the study findings, as well as any compensatory actions during the course of study execution.

Reviewer 3 Report

Comments and Suggestions for Authors

This study built a multidisciplinary team and designed a community-engaged research project to capture community and individual-level exposures to wildfire smoke contaminants, alongside perinatal health outcomes. The following remarks should be taken into consideration preparing updated version of the paper: 1 In table 2, add the question which collection tools are they able to use. This may increase the response rates you mentioned in 3.1. 2 Expect for wildfire, there are many reasons lead to poor infant health, you can add some explains and exclude these effects. 3 In 3.3, Use some long-term monitoring tools to measure the heathy condition and record the results sustainably so that you can compare them. 4 There are only two months “wildfire season”, how does the babies health conditions birth without wildfire season? 5 Does the wildfire effect the infant health, how does it effect and which factor plays the main role? Show your results.
